# What determines the information update rate in echolocating bats

Mor Taub[1,8], Aya Goldshtein [1,5,6,7,8], Arjan Boonman[1], Ofri Eitan[1], Edward Hurme[2,3], Stefan Greif[1] & Yossi Yovel [1,4 ✉]

The rate of sensory update is one of the most important parameters of any sensory system. The acquisition rate of most sensory systems is fixed and has been optimized by evolution to the needs of the animal. Echolocating bats have the ability to adjust their sensory update rate which is determined by the intervals between emissions - the inter-pulse intervals (IPI). The IPI is routinely adjusted, but the exact factors driving its regulation are unknown. We use on-board audio recordings to determine how four species of echolocating bats with different foraging strategies regulate their sensory update rate during commute flights. We reveal strong correlations between the IPI and various echolocation and movement parameters. Specifically, the update rate increases when the signals' peak-energy frequency and intensity increases while the update rate decreases when flight speed and altitude increases. We suggest that bats control their information update rate according to the behavioral mode they are engaged in, while always maintaining sensory continuity. Specifically, we suggest that bats apply two modes of attention during commute flights. Our data moreover suggests that bats emit echolocation signals at accurate intervals without the need for external feedback.

[1] School of Zoology, Faculty of Life Sciences, Tel Aviv University, Tel Aviv 6997801, Israel. [2] Department of Migration, Max Planck Institute of Animal Behavior, Radolfzell, Germany. [3] Centre for the Advanced Study of Collective Behaviour, University of Konstanz, Konstanz, Germany. [4] Sagol School of Neuroscience, Tel Aviv University, Tel Aviv 6997801, Israel. [5] Present address: Department of Collective Behaviour, Max Planck Institute of Animal Behaviour, Konstanz 78464, Germany. [6] Present address: Centre for the Advanced Study of Collective Behaviour, University of Konstanz, Konstanz, Germany. [7] Present address: Department of Biology, University of Konstanz, Konstanz, Germany. [8] These authors contributed equally: Mor Taub, Aya Goldshtein. ✉email: yossiyovel@gmail.com

Animals differ in the rate at which they acquire information about their environment. This has mostly been studied in the visual domain where flicker-fusion measurements reveal large inter-species differences in visual update rates in accordance with their behavior[1–4]. For instance, the pied fly-catcher, an agile bird, exhibits very high visual update rates (138 Hz)[5] while domestic chickens exhibit much lower update rates (87 Hz)[6]. Sampling behavior has also been recorded and studied in additional sensory systems such as whisking[7], touch[8], and electrolocation[9]. Bats gather information about their sur-roundings by actively sampling their environment using ultra-sonic signals. Controlling the properties of the emitted signals such as their frequency, intensity, duration, repetition-rate, and directionality allows bats to refine the acquisition process depending on the task[10]. Searching for prey, prey capture or commuting to and from foraging sites may require different levels of information that can be adjusted by actively changing the update rate at which the environment is sampled. However, it is still not clear how bats determine how often to sample the world. Sparse sampling can lead to loss of relevant information (i.e., obstacles or prey) while over-sampling may lead to expenditure of excess energy. This is especially true because bats couple their emission to the wing-beat in order to save energy[11] and breaking this coupling might be costly[12].

Various models have been suggested for explaining how bats regulate their inter-pulse-interval (IPI) to optimize the rate of information acquisition[13]. One leading hypothesis suggests that bats always maintain an overlap between the sensory volumes covered by consecutive emissions[13,14]. The maximal distance to which an animal can detect a specific object (e.g., prey) is referred to as the sensory volume[15,16] and may have ecological and behavioral implications affecting orientation, obstacle avoidance and hunting. In echolocation, each emission covers a sensory volume that positively depends on the intensity of the emitted signal and might depend on other echolocation parameters in more complex ways. Depending on its flight speed, the bat will travel a certain distance between each two signals, and if this distance exceeds the sensory volume of the echolocation emission, there will be a 'dead-zone' from which the bat will not receive any information. To test the continuity hypothesis, we examined the effect of several echolocation and movement parameters on the IPI: the signal's peak intensity and peak-energy frequency (i.e., the frequency with maximum energy), and flight speed and altitude. Because it is known that bats adjust their IPI to back-ground echoes[17,18], and because the ground is the main reflector during commute, we also tested the effect of flight altitude above the ground. In addition, we tested how bats time their echo-location emission, and specifically, how they emit echolocation signals at constant and accurate intervals and whether they require external feedback to do so.

We used on-board miniature GPS devices with an ultrasonic microphone to track and record the echolocation and flight behavior of four bat species that differ in their foraging habits: (1) the greater mouse-tailed bat (*Rhinopoma microphyllum*), an aerial-hawker that forages in open spaces at relatively high altitudes[19,20], (2) the greater mouse-eared bat (*Myotis myotis*), a ground-gleaning bat which can also hunt on the wing[21], (3) the Mexican fish-eating bat (*Myotis vivesi*), a bat that specializes in hunting small oceanic fish and crustaceans above water at low altitudes[22] and, (4) the lesser long-nosed bat (*Leptonycteris yerbabuenae*) a nectarivorous bat that flies long distances to reach its feeding areas[23,24]. Using on-board recordings allowed us to study sensory regulation in naturally behaving bats and continuously follow the same individuals along the night, thus monitoring how they regulate the sensory update rate over long time periods.

## Results

We recorded the echolocation and movement of four bat species during commute flights, using on-board microphones (foraging flights were excluded from the analysis, see Methods). We termed these flights 'commute' because of their direct nature, but some of the species we studied might have searched for prey during these flights. Adjusting the IPI when foraging in confined space has been studied extensively[25–28], here we focused on commute flights because they are straight and without nearby obstacles, and thus allow to examine the rate of information acquisition in relation to movement (Supplementary Fig. 1). All bats had a similar IPI distribution with a main lobe around 100–150 ms, accounting for an echolocation rate of 7–10 Hz, that is, one emission per wingbeat[29] and a second IPI lobe around 200–300 ms, accounting for flight with one emission every other wingbeat[13,29] (Fig. 1 and Supplementary Fig. 2). As expected from the literature[29,30], the slightly smaller species (*L. yerbabuenae*) exhibited a higher wingbeat rate than the other three that are very similar in size, as could be estimated from the IPIs. We tested the effect of four parameters (signal intensity, peak-energy frequency, flight speed and altitude) on the IPI using Mixed-effect general-ized linear models (GLMMs). The statistical model that best described the data was selected using the Bayesian information criterion (BIC, see Methods). This model revealed a significant effect of all four parameters on the IPI ($P < 0.01$ for the intensity, frequency, speed, and altitude, which were set as fixed factors together with the bat species, while the individual bat's ID was set as a random effect, $n = 22$ bats from all four species in total). In order to compare the relative importance of these factors, we standardized the statistical estimates and compared the direction and weight of each parameter's effect (see Methods, Table 1, and Supplementary Table 1). The most important parameter influ-encing the IPI was the signal's peak-energy frequency. An increase in frequency lead to a decrease in IPI with a decrease in duration of 7.2 ms for every 1 kHz increase in emission frequency. Signal intensity was also negatively correlated to the IPI with a decrease of 2.3 ms in IPI for every 1 dB increase in intensity. The following two parameters, in order of importance, were positively associated with the IPI. The IPI increased with an increase in flight speed and flight altitude (5.4 ms increase in IPI for every $1$ m$\cdot$s$^{-1}$ increase in speed and 0.2 ms increase in IPI for every 1 m increase in altitude). Because there was a significant effect of the bat species on the IPI, we additionally modeled each indivi-dual species (using a GLMM) and found some differences in the importance of the effects of the aforementioned parameters on the IPI for each species (Table 1). In almost all cases, however, the directionality and significance of their effect remained the same, *M. vivesi* being the exception, with a negative effect of altitude on IPI, i.e., as the altitude increased the IPI decreased. This difference might be a result of this bat flying and hunting over water[31] (unlike the others, see Discussion). Note that for some of the species, the best model also included an interaction between speed and intensity (Table 1).

To test the continuity hypothesis, suggesting that bats maintain an overlap between the sensory volumes covered by consecutive emissions, we compared the flight distance the bats travelled between consecutive signals and the sensing range assuming two types of targets: a moth (−40 dB TS at 1 m; Fig. 1c) and a large reflective wall representing a potential obstacle (0 dB TS at 1 m; Supplementary Fig. 3). Three of the four species (all except for *M. myotis*) always conformed to the continuity hypothesis and always adjusted their signal rate such that they never traveled more than the detection range of a moth, thus maintaining a continuous information flow. The fourth species (*M. myotis*) mostly conformed to the continuity hypothesis, except for in a few cases where it exhibited flights with short (up to 50 cm) dead

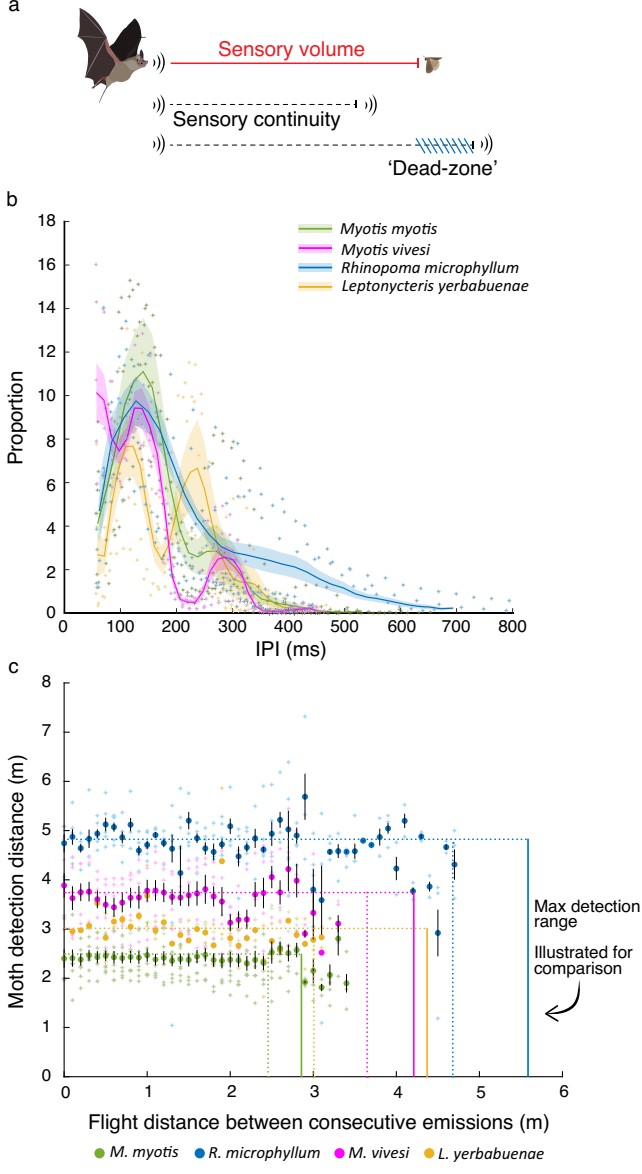

**Fig. 1 Sensing vs. movement. a** Theoretical relationship between the maximum distance in which a bat can detect a moth (sensory volume) and the distance the bat travels between consecutive signals (flight distance). **b** IPI distribution. Lines represent the mean for all conspecifics and shaded areas represent the SE. Asterisks represent the individual data points for each species. $N = 25$ bats from all four species in total (**c**) Moth detection distance (i.e., the 1-Dimensional equivalent of the sensory volume) compared to the flight distance between consecutive emissions of the four bat species. Circles represent the average detection distance for different flight distances based on the actual signal parameters and the flight speed and IPI. Asterisks represent the individual data points for each species. Error bars depict the SE. $N = 22$ bats from all four species in total. Vertical lines show the mean (dotted line) and maximal (solid line) moth detection range for each species. Note that these lines represent values on the Y-axis, but are presented vertically to ease the comparison with the flight distance. These lines reveal that except for *M. myotis* the bats never fly between consecutive signals farther than the detection range thus maintaining sensing continuity for detecting a moth.

zones between emissions (when considering moth detection). When considering a larger target (like a wall) all bats were calling much more than needed in terms of continuity. We note that there was no correlation between flight speed and detection range suggesting that obtaining continuity was not the aim of bats adjustments of echolocation signals.

We next focused on the timing mechanism which allows bats to emit echolocation signals with nearly constant intervals between them. Specifically, we first examined whether bats require external input (i.e., echoes) to time their clock. We hypothesized that if the bats were using external feedback their IPI distribution would be wider (i.e., less precise) when flying high above the ground's detection range, i.e., where echoes are not available in comparison to when they can detect the ground. We tested this in two bat species that flew high enough above ground: for *R. microphyllum* we compared flights at 10–40 m altitude above ground to flights over 90 m and for *M. myotis* we compared flights between 10–20 m to those above 50 m (see Methods regarding ground detection range and flight altitude estimations). There was no significant difference in the IPI distribution width (i.e., the precision) between the two altitude ranges (Supplementary Fig. 4; for *R. microphyllum*: 10–40 m: $100 \pm 80$ ms, mean ± SD vs. over 90 m: $100 \pm 50$ ms, paired Wilcoxon, $P = 1$, $n = 3$ bats. Total number of calls analyzed: 1161. For *M. myotis*: 10–20 m: $70 \pm 20$ ms, mean ± SD vs. over 50 m: $70 \pm 30$ ms, paired Wilcoxon, $P = 1$, $n = 4$ bats. Total number of calls analyzed: 1515), indicating that these bats do not need feedback for high IPI precision.

Finally, we aimed to deepen our understanding of the emission timing mechanism and specifically to distinguish between two alternatives: (a) bat signals are produced with an internal pattern generator (i.e., an internal metronome) in which case there would be no correlation between the time errors of consecutive signals or (b) each signal sets a 'timer' which determines the time of the next signal in which case the time-errors of consecutive signals should be correlated (Fig. 2a, see Methods). We found no correlation between consecutive errors (Pearson correlation, *L. yerbabuenae*: $r^2 = 0.04$, $n = 5$; *M. myotis*: $r^2 = 0.015$, $n = 8$; *M. vivesi*: $r^2 = 0.03$, $n = 6$; *R. microphyllum*: $r^2 = 0.006$, $n = 6$; $P < 0.0001$ for all species. The significant *p*-values result from the large data set, but the variance explained is minimal, Fig. 2b, c).

## Discussion

The rate at which the environment is sampled is a key component of all sensory systems. Depending on the behavioral characteristics of the species, an appropriate flow of sensory information can facilitate obstacle avoidance and prey detection, and reduce the risks of unpredictability. Some species can regulate their sensory acquisition rate. For echolocating bats, regulating this sampling rate can mean the difference between under sampling at the cost of creating a 'dead-zone' and missing relevant information or over-sampling at the cost of decoupling call emission from wingbeat and increasing energetic costs, and at the risk of receiving ambiguous echoes[32]. The factors driving the regulation of the update rate in bats are unknown and, in this study, we use on-board recordings to reveal new insight about them. We analyzed sensory behavior from four species that greatly differ in their foraging strategies: *L. yerbabuenae* commute from their roost to the cacti fields, *R. microphyllum* fly while searching for prey often at high altitudes, *M. myotis* commute to the foraging sites as well, but occasionally also search for aerial insects at low altitudes and *M. vivesi* search for fish while commuting at very low altitudes. Despite these differences the bats seemed to apply the same sensory update mechanisms when controlling their IPI.

**Table 1 P-values and standardized estimates of the statistical models.**

| Species | Parameter | Rank | P-value | Estimate (ms) | Standardized estimate | BIC |
|---|---|---|---|---|---|---|
| *Leptonycteris yerbabuenae* | Peak-energy frequency (kHz) | 2 | **<0.0001** | −3.5 | −0.32 | 6111 |
| | Intensity (dB) | 3 | 0.1 | −1.2 | −0.13 | |
| | Speed (m·s⁻¹) | 4 | 0.2 | −1.1 | −0.06 | |
| | Altitude (m) | 1 | **<0.0001** | 0.3 | 0.37 | |
| *Myotis myotis* | Peak-energy frequency | 1 | **<0.0001** | −6.0 | −0.46 | 132,194 |
| | Intensity | 3 | **<0.0001** | −0.7 | −0.05 | |
| | Speed | 4 | **<0.0001** | 1.7 | 0.04 | |
| | Altitude | 2 | **<0.0001** | 0.9 | 0.23 | |
| | Intensity·speed | | **<0.0001** | 0.4 | | |
| *Myotis vivesi* | Peak-energy frequency | 3 | **<0.0001** | −4.9 | −0.11 | 123,448 |
| | Intensity | 2 | **<0.0001** | −1.9 | −0.21 | |
| | Speed | 1 | **<0.0001** | 12.6 | 0.27 | |
| | Altitude | 4 | **<0.0001** | −0.4 | −0.04 | |
| *Rhinopoma microphyllum* | Peak-energy frequency | 1 | **<0.0001** | −115.4 | −0.67 | 109,180 |
| | Intensity | 2 | **<0.0001** | −4 | −0.14 | |
| | Speed | 3 | **0.004** | 1.7 | 0.03 | |
| | Altitude | 4 | 0.6 | 0.015 | 0.01 | |
| | Intensity·speed | | **<0.0001** | −0.5 | | |
| All species | Species | | **0.0018** | | | 381,942 |
| | Peak-energy frequency | 1 | **<0.0001** | −7.2 | −0.3 | |
| | Intensity | 2 | **<0.0001** | −2.3 | −0.2 | |
| | Speed | 3 | **<0.0001** | 5.4 | 0.15 | |
| | Altitude | 4 | **<0.0001** | 0.2 | 0.1 | |

The standardized estimates were calculated by multiplying the estimate of each parameter with its standard deviation and dividing by the IPIs standard deviation. For each species the most compatible model was selected according to the model's BIC (either with or without an interaction between intensity and speed). The importance of the parameters is ranked from 1–4 (1 being the most important). Significant p-values are marked in bold.

Four echolocation and movement parameters showed a significant correlation with the IPI: the signal's peak-energy frequency and intensity had a negative association, while flight speed and altitude had a positive association (Supplementary Table 1). From the echolocation parameters, the most profound parameter correlating with IPI (except for in *M. vivesi*) was the signal's peak-energy frequency. The negative relationship between these two parameters is well documented for echolocating bats, where signal durations and IPIs are known to decrease when the signal's frequency increases[33–36]. Signal frequency plays a role in determining an object's detection distance. The frequency's effect on the detection range is complex because, on the one hand higher frequencies increase the reflectivity of small targets (i.e., increase target strength[37]) but on the other hand, high frequency signals experience strong atmospheric attenuation[16] and thus might reduce the detection range. Overall, for the frequencies of our bats, increasing the frequency would result in a slight increase in the detection range for small (insect) targets due to higher reflectivity for shorter wavelengths[38]. The update rate also exhibited a significant positive correlation with signal intensity (IPI decreased when echolocation signals were more intense) and three bat species significantly decreased the update rate when flying faster (in *L. yerbabuenae* there was no significant correlation between speed and IPI). In general, all bats conformed to the continuity hypothesis, i.e., they maintained an overlap between the sensory volumes covered by consecutive emissions. The rare occasions where *M. myotis* flew with very short ~50 cm sensory 'dead-zones' could probably be considered errors in our parameter-estimation. In terms of large objects, such as obstacles (which have a longer detection range) all bats always maintained continuity. However, we did not find any correlation between the distance traveled between consecutive calls and the sensory volume (see Fig. 1c). We thus suggest that bats do maintain continuity, but that this is not what drives their adjustments of echolocation and that other factors such as flight altitude and behavioral mode (attentive vs. non-attentive; see below)

determine the sensory update rate. Indeed, many of their adjustments seemed opposite from maintaining continuity. Bats increased their update rate both when increasing the signal's intensity and frequency. Increasing signal intensity or frequency increases the sensory volume (for the frequencies used by these bats[38]) and thus could allow lowering the sensory update rate if the goal was to maintain a fixed sensory volume (i.e., to maintain continuity), however we see the opposite result. Moreover, in contrast with the continuity hypothesis, the bats actually reduced acquisition rate when flying faster. We suggest that the bats engaged in two different modes during their commute, that require different levels of attentiveness. Both tasks are characterized by straight flights (which we characterized as commute flights) but while the low-attention mode is characterized by faster flight, low information update rate and weaker emissions at lower frequencies, the high attention mode is characterized by slower flight speed and a higher information update rate. We hypothesize that this high attention mode is related to searching for prey or attending potential obstacles such as background, conspecifics or the ground. *L. yerbabuenae* differed from the other bats and did not show any correlation between IPI and intensity or speed. This is reasonable when considering that it is the only species that probably exhibited one pure commute mode as it does not catch insects on the wing.

Emitting more intense signals when using shorter IPIs could also play a part in social behavior. *M. vivesi* and *R. microphyllum* rely on unpredictable resources and often commute with many nearby conspecifics[31,39]. In these two species, signal intensity was the second most important parameter affecting the IPI. It is possible that an outcome of their social behavior, is having to amplify echolocation signals in the presence of conspecifics, as was previously shown during foraging in a confined space[28]. The two other species that do not closely commute with conspecifics[31,39] showed a weaker (*M. myotis*) or no correlation (*L. yerbabuenae*) between intensity and IPI. The final parameter affecting the update rate was altitude which was positively

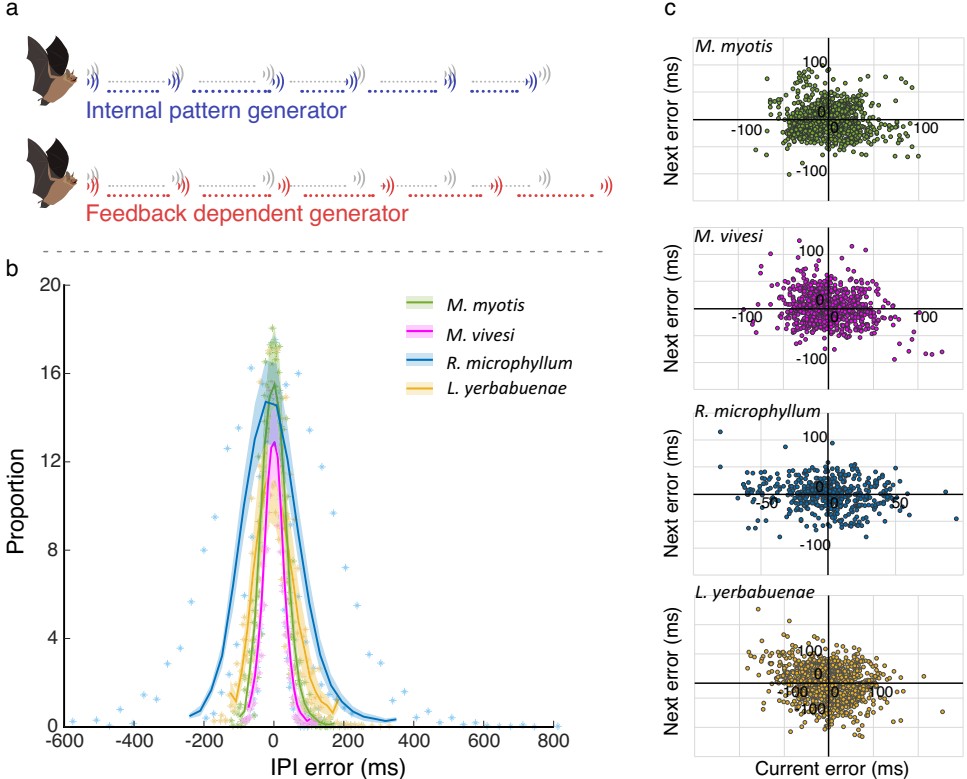

**Fig. 2 IPI timing control. a** Two IPI timing paradigms: The Blue line represents an internal pattern generator while the red represents a feedback dependent generator that relies on the timing set by the previous signal and thus constantly increases over time. Both are compared to the ideal no error grey line. **b** The IPI error distribution for the four bat species. Lines represent the mean for all conspecifics and shaded areas represent the SE. Asterisks represent the individual data points for each species. $N = 25$ bats from all four species in total. **c** Examining consecutive signal error correlations - an example is given for one individual from each species.

correlated with the IPI. The IPI increased with an increase in flight speed and altitude. The decrease in update rate with altitude is well documented[33,40] and probably is a result of less need for information update when there is less clutter (i.e., fewer ground echoes). Relative to the other bats, *M. vivesi* was exceptional showing a negative correlation between altitude and IPI. This discrepancy might be a result of the low maximum flight altitudes used by *M. vivesi* relative to the other species (with a maximum flight altitude of 47 m vs. $329 \pm 215$ (mean ± SD) m for the other three species. Supplementary Fig. 5). Scandinavian bats that migrate and commute above the sea were found to fly at low altitudes, presumably to allow them to orient relative to the water surface[41]. In line with the results obtained in this study for *M. vivesi*, the Scandinavian bats used lower signal frequencies and longer IPIs at low altitudes. *M. vivesi* hunts for oceanic prey in the water surface, a food resource that is unpredictable in both time and space[42]. Lower flight altitudes might allow them to search for unpredictable patches of food along the commute route, therefore their limited flight range does not have the same association with the sample rate.

To maintain a constant update rate, bats must accurately estimate short time periods. In accordance with previous studies[17,18,43,44], we found that bats adjust their IPI based on altitude at low heights (<20–40 m depending on the species), but that the IPI precision did not depend on receiving ground echoes. Many bat species sometimes fly far from the ground, where they cannot get echoic feedback. We show that bats use an internal timing mechanism to control the timing of emissions. Previous studies have already shown that bats can use internal cues to regulate sensing. For instance, Doppler shift compensation

behavior was found to be controlled by internal cues when external echoic feedback was not available[45]. In conclusion, by recording audio on-board four bat species which greatly differ in their foraging characteristics, we found that similar factors probably act to determine their information update rate which overall facilitates continuous sensory acquisition but is probably also adapted to the mode of behavior.

## Methods

**Animals**. The experimental protocols and procedures were approved and performed according to the Institutional Animal Care and Use Committee of the Israeli Health Ministry, permit #L-11-054. All animal capture and experiments were conducted according to the following permits of the responsible authorities: *R. microphyllum* (Israel): permits #2011/38346 and 2012/38346 from the NPA (Israel Nature and Parks Authority); *L. yerba-buenae* (Mexico): permit #04019/15, 03946/15 14509/16 from Dirección General de Vida Silvestre (General directorate of Wildlife); *M. myotis* (Bulgaria): MOEW-Sofia (Ministry of Environment and Water) and RIOSV-Ruse (Regional Inspectorate of Environment and Water), Bulgaria, permit #465/ 29.06.2012 and 639/28.05.2015. *M. vivesi* (Mexico): permits #7668-15 and 2492-17 from Dirección General de Vida Silvestre, and permits # 17-16 and 21-17 from Secretaría de Gobernación (Secretariat of the Interior). We have complied with all relevant ethical regulations for animal use.

**Tracking bats' movement and echolocation**. The movement and echolocation data of all four species was already published

**Table 2 Detailed information about the tracked bats.**

| Species | Sample size | Tracking device | Audio sample rate (Hz) | GPS sample rate (Hz) | Sex and reproductive state | Flight duration (of analyzed calls, minutes) | Flight distance (of analyzed calls, km) | Number of calls[a] |
|---|---|---|---|---|---|---|---|---|
| *Leptonycteris yerbabuenae* | 2 | Robin | 187,500 | 0.067 | Lactating females | 16.2 ± 12.7 | 10.3 ± 12.1 | 1144 ± 962 |
|  | 3 | Vesper | 200,000 | NaN |  |  |  |  |
|  | 1 |  | 187,500 | 0.067 | Post-lactating females | 29.4 ± 14.0 | 10.5 ± 4.6 | 1519 ± 462 |
| *Myotis myotis* | 1 | Robin | 375,000 | 0.067 |  |  |  |  |
|  | 6 |  | 375,000 | 0.2 |  |  |  |  |
| *Myotis vivesi* | 6 | Robin | 187,500 | 0.067 | Lactating females | 76.2 ± 35.5 | 24.4 ± 10.0 | 1819 ± 1195 |
| *Rhinopoma microphyllum* | 6 | Robin | 93,750 | 0.067 | Post-lactating females | 49.8 ± 18.5 | 16.1 ± 5.1 | 1442 ± 1591 |

Note that three of the five *L. yerbabuenae* did not have speed and altitude data and were therefore excluded from analyses that included these parameters (GLM models and sensory volume figures).
[a]Number of calls represents the number of analyzed calls (calls during commute flight with IPI > 0.05)

elsewhere with detailed methods[23,31,39,42]. We will thus briefly describe them here. To examine how bats regulate their inter-pulse intervals, we tracked the flight behavior and echolocation of four species from three different families (Table 2): *Leptonycteris yerbabuenae* (Phyllostomidae), *Myotis myotis* (Vespertilionidae), *Myotis vivesi* (Vespertilionidae), and *Rhinopoma microphyllum* (Rhinopomatidae).

Bats were captured next to their roost using a mist net, hand net, or gloved hand. We tracked bats' movement and echolocation using a miniature GPS device (Robin, Lucid Ltd., Israel or Vesper, ASD inc., Israel) with an on-board ultrasonic microphone (FG-23329, Knowles) that was attached to a telemetry unit (LB-2 × 0.3 g, Holohil Systems Ltd., Canada). The devices were wrapped with waterproof balloons and were mounted on the back of the bats using surgical cement (Perma-Type, McKesson Patient Care Solutions, Inc., USA). The bats were held for a few minutes to allow the glue to dry properly, then rested for another 15 min in a fabric bag and released at the same location. The devices were located using telemetry a few days later, and retrieved after the device fell off the bat or gently removed from the recaptured bat (the devices remained on the bats for ~3 days). The total mass of the tracking device was 4.3 g on average and reached a maximum of 15% of the body mass of the bats (with an average of 14% for *M. myotis*, 13.8% for *L. yerbabuenae*, 12% for *R. microphyllum*, and 14% for *M. vivesi*; see more information about the tagging process of each species, control experiments and detailed discussion about the effect of the extra weight of the GPS device on bats' flight behavior[23,31,39]).

**Movement analysis**. Prior to the GPS data analysis, outliers were deleted using the dilution of precision index and ground speed. Afterwards, the data was interpolated linearly and smoothed using 'LOESS' local regression smoothing filter[46]. The raw GPS data included time, longitude, latitude, and altitude above the ellipsoid. Bat ground speed between consecutive GPS points was calculated. The altitude above the surface was calculated by subtracting the geoid and the elevation above the surface from the ellipsoid altitude. The geoid height was estimated to an accuracy of 0.001 m using EGM2008 Geopotential Model, and the surface elevation was extracted using Google Maps Elevation API. We relied on the GPS measurements to estimate flight altitude. GPS-based altitude estimates are inaccurate with errors of −5 ± 11 m (mean ± SD) in our case[39]. However, for the timing precision analysis, we took a large margin between the two groups (larger than the error). Moreover, this error would introduce error to the GLM analysis, but not a bias so any effect that is found is probably even stronger.

Bat trajectories were divided into commute and foraging segments using a straightness index[47]. For each GPS point, we measured the ratio between the length of the beeline and the actual flight path of long and short flight segments. This allowed assessing the index in two scales to make sure we recognize very local foraging behavior in addition to the main foraging sites of each bat. Points with a straightness index greater than a certain threshold for the short or long segments were considered as commute and all other points were considered as foraging. The long and short segment lengths and the thresholds were defined according to the flight behavior of each individual based on a binomial distribution of their straightness index (short segments were defined by 6–10 data points and a threshold value of 0.35–0.65, and long segments were defined by 30–120 data points and a threshold of 0.31–0.8). All foraging data was then excluded from the analysis and any further analysis was conducted only for flight and echolocation during commute flights.

**Audio analysis**. Audio files were synchronized to the GPS data, and were tagged to the nearest GPS sample. The microphones of the tracking devices were calibrated, meaning that for each frequency received, voltage could be converted into dB SPL. During calibration a reference frequency sweep (covering the range of all species in the study) was recorded by both the tracking device's microphone and a calibrated instrumentation microphone (GRAS DP40 1/8") so that the input signal was known in dB SPL at each frequency. Echolocation analysis included signals peak-energy frequency - the frequency with the highest energy in the spectrum, signal intensity (peak energy), signal duration, and inter-pulse intervals of the echolocation signals during commute flight. Due to the limited battery life and memory capacity of the recording devices, we recorded a 0.5 s audio window every 5 s (with two *R. microphyllum* bats having 5 s every 30 s recordings), limiting the IPI analyzed. The signal to noise ratio differed strongly between different bat species because they occupy different frequency bands at which the tracking device's microphone is more or less sensitive. We cut out each call just above the local noise level around the call defined by trial and error. This method ensured that we maximized the information use in each recording (from the cut out calls we can measure the S/N ratio retrospectively). In addition, we used the sonar equation[37,48] to calculate the maximum distance from which a bat can detect a moth (wing surface: 200 mm × 2; −40 dB TS at 1 m[49]) and a large reflective wall (0 dB TS at 1 m). For all targets −6.02 dB loss per doubling of distance was assumed and atmospheric attenuation was modelled according to[50–52]. Audio analysis was conducted using Matlab.

**Statistics and reproducibility**. We tested the effect of four parameters on the IPI using Mixed-effect generalized linear models (GLMMs): the signal's peak intensity, peak-energy frequency, flight speed and flight altitude. The statistical model that best described the data was selected using the Bayesian information criterion (BIC) after testing additional models including pairwise interactions between variables.

The standardized estimates were calculated by multiplying the estimate of each parameter with its standard deviation and dividing by the IPIs standard deviation[53]. The actual effect of the different parameters on the IPI was derived from the original estimates of the model and described as the change in IPI (in ms) for one unit of each respective parameter. All statistical analyses were performed in JMP software (SAS institute Inc., USA).

To examine the need for external feedback when timing IPIs, we analyzed the IPI distribution for two of the four species because only these species had sufficient data from a wide enough altitude range. The relevant altitude ranges for each species were estimated by finding the altitude that did not have a significant effect on the IPI using multiple GLM tests for individual bats. To this end we gradually removed data from increasing altitudes and ran a model including the IPI as the response variable, the altitude as a predictor and intensity and peak-energy frequency as covariates. Once there was no effect of altitude on the IPI, we assumed that bats cannot sense the ground. The estimate was more conservative than an acoustic estimate (based on the sonar equation) and thus we preferred to use it and make sure that the bats could not sense the ground. To estimate the IPI distribution width, for each bat, we measured the peak of the IPI histogram and measured the width of the full curve at the half point from the peak. We then ran a paired Wilcoxon test to compare the IPI distribution width between the two altitude ranges for each species.

In order to distinguish between possible timing mechanisms, we examined the correlation between consecutive time errors within signal sequences using the Pearson correlation computed for all sequences of individual bats within each species (by individual). For each sequence we calculated the time error by subtracting the mean IPI from each IPI in the sequence (this assumes a desirable constant IPI). Sequences with fewer than four signals were excluded from the analysis. Since sequences were limited by the sampling duration defined for each species' recordings, long IPIs (longer than 0.5 s) were not included in the analysis (apart from two *R. microphyllum* that had longer segments).

**Reporting summary**. Further information on research design is available in the Nature Portfolio Reporting Summary linked to this article.

## Data availability

The datasets generated and analyzed during the current study are available on Mendeley Data: https://doi.org/10.17632/w4s2xrkv6p.1[54].

## Code availability

The codes used for analysis are available on Mendeley Data: https://doi.org/10.17632/w4s2xrkv6p.1[54].

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

## Acknowledgements

We would like to thank Luis Gerardo Herrera M., José Juan Flores-Martínez, Andrea T. Valdés, Rodrigo A. Medellin, Dave S. Johnston, Katya Egert-Berg, Ivo Borissov, Noam Cvikel, Jeremy Ryan Shipley, Gerald S. Wilkinson, and Holger R. Goerlitz for assistance with data collection. This research was funded by ISF.

## Author contributions

A.G., Y.Y., and M.T. Conceived and designed the research. A.G., E.H., O.E. and S.G. collected the data. M.T., A.G., and A.B. analyzed the data. M.T., A.G. and Y.Y. wrote the manuscript. E.H. and A.B. reviewed and edited the manuscript. All authors read and approved the final manuscript.

## Competing interests

The authors declare no competing interests.
