## [Peer Review File · Communications Biology]

Reviewers' comments:

Reviewer #1 (Remarks to the Author):

This manuscript explores an interesting question in the field of echolocation and auditory processing. Though I found the manuscript generally easy to read and well organized, I have a few comments that I hope can help the authors clarify some of their points.

- The authors use the word acquisition throughout the manuscript and title but it is not clear if they are actually referring to emissions. The word 'acquisition' should be used if flight velocity is integrated into the calculation and therefore the measure refers to echo arrival at ears. If the authors did this, it needs to be spelled out further. The figure legend of Figure 1 A seems to indicate that velocity was taken into account to calculate acquisition but further details on this need to be added to the methods. If the authors are referring to emission rate, then the manuscript needs to be reworded. This is especially confusing in line 217. Needs clarification.
- In line 48 the authors mention the 4 parameters they investigated (which are clear later on in the manuscript) but only mention 3 in the sentence, Altitude is missing.
- I suggest the authors arrange Table 1 in the same order for each species. Relevance of each parameter could be added as a new column but for easy comparison across species, it would be useful if parameters were presented in the same order.
- In line 109 the authors introduce the 'dead zone' flights of myotis bats and discuss it in line 220. I suggest citing other work in bats that reports 'flying in silence' and the potential for eavesdropping on other bats (i.e. Chiu et al., 2008).
- The authors should elaborate further in the discussion on the second lobe of IPIs for *Leptonycteris* and *Myotis* bats at 250ms and 300ms respectively.
- As stated by the authors in lines 212-213, high frequencies attenuate faster, potentially reducing the sensory volume. This is contradicting lines 227-229. This second sentence needs rewording for clarification.
- Table 2 should have a column stating the bats weight range and % in regards to the tag. This is especially concerning considering that for two species the animals used were lactating females that potentially need to carry their pups and are thus already at weight overload. Weight overload may affect maneuverability and flight parameters. These constraints need to be addressed in the manuscript.

Reviewer #2 (Remarks to the Author):

This study analyzes how echolocating bats control the timing of their pulse emissions, based on valuable data measured from wild bats. Studies have already been reported on IPI in a lab and in small-scale field settings, and some of the rules have become well-known. However, in this study, on-board microphones were used to accurately measure IPI and movement in naturally behaving bats with four different foraging habits, which are inherently difficult to measure. This is very valuable data to study the sampling behavior of bats by knowing the relationship between IPI and acoustic parameters, and the results are well described. On the other hand, the statistical focus of the results seemed to mask some of the appeal of the valuable biologging data. In addition, the lack of consideration of the effect of task is another concern of this study, since the analysis is not categorized into commute or search situations (or tasks, echolocation phases,...). It is also unfortunate that the results of four different bat species with different foraging habits are not well discussed in relation to commuting or foraging habits. If the environment in which the bats are flying can be determined from GPS, it would be interesting to see how IPI control relates to the surrounding environment. Specific comments are given below.

L66 A brief explanation of the definitions of "commute" and "search" is needed (Method explained how to classify "commute" and "foraging"). Also, were there any particular differences in the IPI analysis

results between commuting and foraging flights?

L82-84 I understand that "increase in frequency" is a decrease in IPI, but "signal intensity was negatively correlated to the IPI" seemed to me to be the opposite result such as IPI increases with higher sound pressure from previous studies, etc.

Table 1: "2.3 ms in IPI for every 1 dB" in the text is consistent with the value in the All species column of Table 1, but "6.7 ms for every 1 kHz" may refer to 7.2 ms in Table 1? (L86: Is 5.6 ms also a mistake for 5.4 ms?). I may not have understood it correctly. I also wonder if you could show these results as a supplemental figure.

Since these data are already published in other papers, but they are all valuable data, why don't you include a spectrogram of the sound of each bat species measured by the loggers, and also some figures such as showing the position of pulse emission on the flight trajectory, so that we can visually understand the change in IPI, either in the text or as a supplemental figure?

Fig.1B I think the absence of IPI over 500 ms is due to intermittent recordings of 0.5 seconds, but only for *R. microphyllum*, I think the data also include the results of intermittent recordings of 5 seconds. For the distribution of IPI for the three bat species with a recording time limit of 0.5 seconds, is there any effect on the results in Fig. 1C (e.g. *Myotis myotis*)?

Please also provide information such as the number of pulses and audio data length (time and flight distance) used in this analysis, just in case.

It would be interesting to see how IPI control is related to the foraging habits of each bat species and to the surrounding environment, such as the flight route of each individual bat.

L213 The result in the text says "an increase in frequency lead to a decrease in IPI". I think what is written here is the opposite of the result described in the main text.

In terms of acoustics, it seems that the change in the size of objects that can be detected by the increase in frequency observed in this study is almost negligible. As I commented in the results section, generally speaking, as the sound pressure of the transmitted signal increases, the detection distance increases and the IPI also increases, but the actual results were the opposite. I think this may be related to the fact that the commute and search situations are not analyzed separately, so I suggest that you re-analyze the analysis with this in mind. The possibility of social behavior is also mentioned in the discussion that follows, but again, this is unknown from the results. Regarding the question of what determines the IPI of bats, the readers would like to know more about the reasons for the relationship with sound pressure, which had a significant parameter.

Fig.2A I think the order of the explanations in the text and figure captions does not match the figures.

P354 When the effect of altitude on IPI was statistically observed, was the altitude at which the bats were flying lower than the altitude at which the echoes returned from the ground? A graph showing the change in altitude and IPI over time would be helpful here as well.

Reviewer #3 (Remarks to the Author):

Taub et al. What determines the information acquisition rate in echolocating bats.
Communications Biology manuscript COMMSBIO-23-2814

The manuscript tackles the question how echolocating bats adjust sensory acquisition by controlling different properties of the emitted signals. To this behalf, the authors re-analysed already published

data sets (see Methods section, line 278) acquired with on-board audio-recordings in combination with GPS tracking in 4 different bats species during commute and search flights. Using mixed-effect generalized linear models (GLMMs), the effect of four echolocation and movement parameters on the sensory acquisition rate, i.e. the inter pulse interval (IPI) of the echolocation calls, was investigated. The results suggest, that bats maintain sensory continuity by dynamically controlling acquisition rate according to the actual behavioral mode.

Comments

Major

The manuscript is generally very well written and the described methods are technically sound. The results are new and interesting for scientists working in the field of sensory ecology. My main concern, however, is that the interpretation of the results from GLMMs is not fully comprehensible. In the following I will explain this in more detail:

The authors claim in the discussion that the bats maintain sensory continuity. However, the effect of some of the investigated parameters on IPI was contradictive to that. For example, flight speed had a negative effect on acquisition rate, a finding that is not in accordance with the sensory continuity hypothesis. The authors admit that in the discussion (line 224-234), however, they try to resolve this contradiction by assuming that the behavioral mode (commute or search flight) has a strong influence on the sensory acquisition rate. And I think the authors are basically right to assume that. However, the important point is that they cannot prove this from their data, I think. According to the information given in the methods section the analysis of bat trajectories acquired by GPS measurements only allowed for discrimination of commute and foraging flight based on the "straightness index" (line 313-315). However, in the last sentence of the paragraph (line 318-320) the authors claim that "... any further analysis was conducted only for flight and echolocation during search and commute flights". How was the distinction made between these two behavioral modes? As long as data for search and commute flight cannot be analyzed separately it is a bit vague to hypothesize that the bats basically adjust the acquisition according to the sensory continuity hypothesis while contradictive data is explained by the (in fact unknown) behavioral modes.

In addition, it also seems a bit futile to state that in "... four bat species which greatly differ in their foraging characteristic (...) similar factors probably act to determine their information update rate..." (Discussion, line 264-2366), if only data was analyzed which was recorded during commute flight, which is probably not very different in all species investigated in the study.

Minor:

Methods, line 335, it should read 0 dB TS... , I guess.

Reviewer #1:

1. The authors use the word acquisition throughout the manuscript and title but it is not clear if they are actually referring to emissions. The word 'acquisition' should be used if flight velocity is integrated into the calculation and therefore the measure refers to echo arrival at ears. If the authors did this, it needs to be spelled out further. The figure legend of Figure 1 A seems to indicate that velocity was taken into account to calculate acquisition but further details on this need to be added to the methods. If the authors are referring to emission rate, then the manuscript needs to be reworded. This is especially confusing in line 217. Needs clarification.

The term 'information acquisition' is widely used to describe controlled active sensing (Nelson & Maclver, 2006) which is why we used it throughout the manuscript even though only some of the analysis takes flight velocity into account. We now changed most of the phrasing to 'sensory update rate'. We feel that 'emission rate' doesn't fully capture the intent by only describing the action but not the function.

2. In line 48 the authors mention the 4 parameters they investigated (which are clear later on in the manuscript) but only mention 3 in the sentence, Altitude is missing.

The next sentence refers to the altitude in more detail, which is why it was separated from the rest of the parameters in the sentence. We rephrased for clarity (lines 49-53).

3. I suggest the authors arrange Table 1 in the same order for each species. Relevance of each parameter could be added as a new column but for easy comparison across species, it would be useful if parameters were presented in the same order.

We rearranged the order of parameters according to this suggestion and added a column describing the importance.

4. In line 109 the authors introduce the 'dead zone' flights of myotis bats and discuss it in line 220. I suggest citing other work in bats that reports 'flying in silence' and the potential for eavesdropping on other bats (i.e. Chiu et al., 2008).

We now refer to the 2008 study in the beginning of the paper as an example for other types of IPI control during foraging (line 70). However, we would like to note the 'dead zone' that we refer to is not caused because the bats stop calling, but instead because they are flying faster than their update rate. In addition, in the 2008 study, flying in silence was observed for short periods of up to 200ms during foraging while our study focuses on commute flights.

5. The authors should elaborate further in the discussion on the second lobe of IPIs for Leptonycteris and Myotis bats at 250ms and 300ms respectively.

The second IPI lobe is a result of the bats calling every other wingbeat and has been described in the past (Jones, 1994; Holderied & von Helversen, 2003). We added an additional description of the second lobe in the results (lines 74-75).

6. As stated by the authors in lines 212-213, high frequencies attenuate faster, potentially reducing the sensory volume. This is contradicting lines 227-229. This second sentence needs rewording for clarification.

The effect of frequency is dual due to the attenuation on the one hand and the effect on target strength on the other hand. For the frequencies used by our bats, increasing frequency would slightly increase the sensory volume (as can be seen clearly in the graph in Houston et al., 1999). We state this more clearly (now lines 248-249). This is in spite of the effect of attenuation discussed in lines 212-213 (now 246-247). We later state that this result could allow the bats to decrease the sensory update rate if the goal was to maintain a fixed sensory volume, but in fact we see the opposite result (262-264). We rephrased these lines in the manuscript for clarity.

7. Table 2 should have a column stating the bats weight range and % in regards to the tag. This is especially concerning considering that for two species the animals used were lactating females that potentially need to carry their pups and are thus already at weight overload. Weight overload may affect maneuverability and flight parameters. These constraints need to be addressed in the manuscript.

We now added the specific % per species (lines 330-331). The data used in this manuscript was described in length in previous publications (see Egert-Berg et al., 2018) where there is a detailed discussion of the effect of the extra weight including several control experiments examining for example endurance and foraging. The decision to tag lactating females was discussed with nature authorities and was agreed to reduce the stress on the population because it allows tagging very few bats for very short periods. We note that our tags weigh more or less like a new-born pup which the mothers carried in uterus and have to carry occasionally (the species we worked with - *Leptonycteris yerbabuenae* and *Myotis Vivesi* - do not forage with pups on-board. The pups stayed alone at the roost while the mothers forage alone (Goldshtein et al., 2020, and Egert-Berg et al., 2018)). We refer the readers to that discussion in lines 330-332 of the manuscript ("see more information about the tagging process of each species, control experiments and detailed discussion about the effect of the extra weight of the GPS device on bats' flight behavior 23,27,35.").

Reviewer #2:

The statistical focus of the results seemed to mask some of the appeal of the valuable bio logging data. In addition, the lack of consideration of the effect of task is another concern of this study, since the analysis is not categorized into commute or search situations (or tasks, echolocation phases). It is also unfortunate that the results of four different bat species with different foraging habits are not well discussed in relation to commuting or foraging habits. If the environment in which the bats are flying can be determined from GPS, it would be interesting to see how IPI control relates to the surrounding environment. Specific comments are given below.

We would like to clarify (and we now do so in the manuscript as well, lines 68-71) that the searching mode that we refer to throughout the manuscript is happening during direct flights

which we term commute. We did not analyze search flights *that occur during foraging*. This does not mean that the bats are not searching during the commute. We now refer to these modes as more or less attentive as we clarify in the manuscript (lines 268-274).

We believe that studying information acquisition rate from a sensorimotor point of view is easier during these flights for several reasons. When bats are flying in cluttered environment near background it is well documented that the distance to the background is the main factor affecting IPI. As we cannot accurately reconstruct the distance to background from GPS data, we would suffer from a large error. IPI adjustments are better researched in confined rooms (or using LIDAR to reconstruct small spaces), while we could only provide rough estimates. But studying information acquisition rate in direct flight with little background (except for the ground sometimes) allows examining how changes in movement, and probably also in task, affect information acquisition rate.

This is also why the different continuity hypotheses have been discussed in reference to such 'commute' flights.

1. L66 A brief explanation of the definitions of "commute" and "search" is needed (Method explained how to classify "commute" and "foraging"). Also, were there any particular differences in the IPI analysis results between commuting and foraging flights?

Good point. We now added a clarification about the use of "search" as a part of the commute phase and not the foraging phase (lines 68-71, 268-274). We did not analyze foraging flights in this work.

2. L82-84 I understand that "increase in frequency" is a decrease in IPI, but "signal intensity was negatively correlated to the IPI" seemed to me to be the opposite result such as IPI increases with higher sound pressure from previous studies, etc.

Our results are indeed not in line with all previous studies, however past work mostly measured this relationship during foraging and not during commute. One study that looked at the effects of acute interference during search and approach did however find that bats emitted calls of higher intensity while calling more often (thus reducing their IPI) when there were other bats around (Amichai et al., 2015). We added this example to the discussion (line 279). In any case, this was the result for four species, suggesting that we are not looking at some artifact.

3. Table 1: "2.3 ms in IPI for every 1 dB" in the text is consistent with the value in the All species column of Table 1, but "6.7 ms for every 1 kHz" may refer to 7.2 ms in Table 1? (L86: Is 5.6 ms also a mistake for 5.4 ms?). I may not have understood it correctly. I also wonder if you could show these results as a supplemental figure.

We corrected the text to describe the non-standardized estimates that appear in Table 1.

4. Since these data are already published in other papers, but they are all valuable data, why don't you include a spectrogram of the sound of each bat species measured by the loggers, and also some figures such as showing the position of pulse emission on the flight trajectory, so that we can visually understand the change in IPI, either in the text or as a supplemental figure?

We thank the reviewer for this comment and have now added a supplementary figure (Supplementary Fig. 1) of the bats flight trajectory as well as a spectrogram of the corresponding signals for one individual from each species.

5. Fig.1B I think the absence of IPI over 500 ms is due to intermittent recordings of 0.5 seconds, but only for *R. microphyllum*, I think the data also include the results of intermittent recordings of 5 seconds. For the distribution of IPI for the three bat species with a recording time limit of 0.5 seconds, is there any effect on the results in Fig. 1C (e.g. *Myotis myotis*)? Please also provide information such as the number of pulses and audio data length (time and flight distance) used in this analysis, just in case.

This is correct, we cannot account for IPIs that are longer than 0.5 s because of the recording definitions. We now added a clear comment about the length of the recordings in the text (lines 382-384) and added the requested information in Table 2.

6. It would be interesting to see how IPI control is related to the foraging habits of each bat species and to the surrounding environment, such as the flight route of each individual bat.

As noted above, it is well documented in the literature that IPI is mostly affected by the vicinity of reflective clutter, but because we cannot determine the bat's exact distance from background elements the momentary IPI would be hard to interpret. This is indeed a very interesting idea for future research with on-board recordings. We also added a supplementary figure depicting the IPI along the flight trajectory with regard to the altitude above ground (Supplementary Fig. 5).

7. L213 The result in the text says "an increase in frequency lead to a decrease in IPI". I think what is written here is the opposite of the result described in the main text. In terms of acoustics, it seems that the change in the size of objects that can be detected by the increase in frequency observed in this study is almost negligible. As I commented in the results section, generally speaking, as the sound pressure of the transmitted signal increases, the detection distance increases and the IPI also increases, but the actual results were the opposite. I think this may be related to the fact that the commute and search situations are not analyzed separately, so I suggest that you re-analyze the analysis with this in mind. The possibility of social behavior is also mentioned in the discussion that follows, but again, this is unknown from the results. Regarding the question of what determines the IPI of bats, the readers would like to know more about the reasons for the relationship with sound pressure, which had a significant parameter.

As we clarify above, the search phase we refer to is not a foraging search (which we do not address in this study). We detect direct flights (in straight lines) which we term commute. Only the echolocation behavior which indeed contradicted our expectations suggests that there might be two different behavioral modes: a high attention mode (perhaps this is a better name than search) where bats call more often and at higher intensities and frequencies, and a low attention mode. We hope that we clarified this in the revised manuscript. We hypothesize that these two modes are equivalent to the situations when the bats either fly fast ignoring any potential background or prey (which we refer to as 'commute') or they fly while 'searching' for prey or attending the background (maybe the ground).

The increase in frequency indeed has a very small effect on the detection range – but it is probably a positive one (see Houston et al., 1999).

8. Fig.2A I think the order of the explanations in the text and figure captions does not match the figures.

The order of the timing paradigms in Fig 2A was switched for clarity.

9. P354 When the effect of altitude on IPI was statistically observed, was the altitude at which the bats were flying lower than the altitude at which the echoes returned from the ground? A graph showing the change in altitude and IPI over time would be helpful here as well.

We are not sure if we understood the question, but we provide a full IPI and altitude profile for each species in a new figure (Supplementary Fig. 5).

Reviewer #3:

1. My main concern, however, is that the interpretation of the results from GLMMs is not fully comprehensible. In the following I will explain this in more detail:

The authors claim in the discussion that the bats maintain sensory continuity. However, the effect of some of the investigated parameters on IPI was contradictive to that. For example, flight speed had a negative effect on acquisition rate, a finding that is not in accordance with the sensory continuity hypothesis. The authors admit that in the discussion (line 224-234), however, they try to resolve this contradiction by assuming that the behavioral mode (commute or search flight) has a strong influence on the sensory acquisition rate. And I think the authors are basically right to assume that. However, the important point is that they cannot proof this from their data, I think. According to the information given in the methods section the analysis of bat trajectories acquired by GPS measurements only allowed for discrimination of commute and foraging flight based on the “straightness index” (line 313-315). However, in the last sentence of the paragraph (line 318 320) the authors claim that “... any further analysis was conducted only for flight and echolocation during search and commute flights”. How was the distinction made between these two behavioral modes? As long as data for search and commute flight cannot be analyzed separately it is a bit vague to hypothesize that the bats basically adjust the acquisition according to the sensory continuity hypothesis while contradictive data is explained by the (in fact unknown) behavioral modes.

We agree with the reviewer. It does not seem that the bats aim to achieve continuity, but that they achieve it anyway. We did not try to argue otherwise, and we now tried to clarify this in the revised manuscript.

We would like to clarify (and now do so in the manuscript as well, lines 68-71) that the **search** mode that we refer to throughout the manuscript is happening during the commute. We did not analyze search flights that occur at foraging sites. Based on our results, and because bats don't seem to aim for continuity, we hypothesized that within what we originally identified as commute there are actually two modes: bats either (1) fly fast ignoring the background (which we refer to as 'commute') or they (2) fly more attentively which we refer to as 'search during commute'. During the more attentive “search” phase the bats call more often, louder and in higher

frequency (violating the continuity hypothesis). We now also refer to these modes as attentive and non-attentive modes which hopefully clarifies our meaning.

We added a distinction between the different modes to the discussion and also clarify that this is only a hypothesis as the reviewer correctly suggests (lines 268-274).

2. In addition, it also seems a bit futile to state that in “ ... four bat species which greatly differ in their foraging characteristic (...) similar factors probably act to determine their information update rate...” (Discussion, line 264-236), if only data was analyzed which was recorded during commute flight, which is probably not very different in all species investigated in the study.

All the species investigated in this study are commuting, however they greatly differ from one another when doing so: *Leptonycteris yerbabuenae* are only moving from one spot to the next, *Rhinopoma microphyllum* are searching for insects, *Myotis vivesi* are searching for fish, and *Myotis myotis* are commuting to the foraging sites but at relatively low altitudes and probably occasionally searching for aerial prey. One could thus hypothesize that they have very different information update needs, but in spite of these differences it seems that they all behave similarly when controlling their IPI. We added this clarification to the discussion (lines 232-237).

3. Methods, line 335, it should read 0 dB TS, I guess.

The error was fixed.

Additional figures that were added to the manuscript:

Supplementary Fig. 1:

Supplementary Figure 1: Examples of bats' flight trajectories and echolocation calls. Bats' flight trajectories of one individual of (A) *L. yerbabuena*, (B) *M. myotis*, (C) *M. vivesi*, and (D) *R. microphyllum* are color-coded according to their ground speed (blue to yellow scale). Black dots represent GPS positions without analyzed audio data. The colonies of the bats are presented by red circles. Inset panels show examples of bats' echolocation calls, with corresponding recording locations marked by purple circles.

Supplementary Fig. 5:

Supplementary Figure 5: Examples of bats' IPI by altitude along the flight trajectories. Bats' flight trajectories and flight altitude above ground (insert panels) of one individual of (A) *L. yerbabuena*, (B) *M. myotis*, (C) *M. vivesi*, and (D) *R. microphyllum* are color-coded according to their IPI (blue to yellow scale). Black dots represent GPS positions without analyzed audio data. The colonies of the bats are presented by red circles.

REVIEWERS' COMMENTS:

Reviewer #1 (Remarks to the Author):

The authors have addressed all my comments.

Reviewer #2 (Remarks to the Author):

The authors have provided additional explanations and other detailed responses in the revised manuscript, so I have no further comment.

Reviewer #3 (Remarks to the Author):

The authors were able to address my concerns appropriately.
The manuscript is now ready for publication.